# HIEROS: HIERARCHICAL IMAGINATION ON STRUCTURED STATE SPACE SEQUENCE WORLD MODELS

## ABSTRACT

One of the biggest challenges to modern deep reinforcement learning (DRL) algorithms is sample efficiency. Many approaches learn a world model in order to train an agent entirely in imagination, eliminating the need for direct environment interaction during training. However, these methods often suffer from either a lack of imagination accuracy, exploration capabilities, or runtime efficiency. We propose HIEROS, a hierarchical policy that learns time abstracted world representations and imagines trajectories at multiple time scales in latent space. HIEROS uses an S5 layer-based world model, which predicts next world states in parallel during training and iteratively during environment interaction. Due to the special properties of S5 layers, our method can train in parallel and predict next world states iteratively during imagination. This allows for more efficient training than RNN-based world models and more efficient imagination than Transformer-based world models. We show that our approach outperforms the state of the art in terms of mean and median normalized human score on the Atari 100k benchmark, and that our proposed world model is able to predict complex dynamics very accurately. We also show that HIEROS displays superior exploration capabilities compared to existing approaches.

## 1 INTRODUCTION

Learning behavior from raw sensory input is a challenging task. Reinforcement learning (RL) is a field of machine learning that aims to solve this problem by learning a policy that maximizes the expected cumulative reward in an environment (Sutton & Barto, 2018). The agent interacts with the environment by taking actions and receiving observations and rewards. The goal of the agent is to learn a policy that maximizes the expected cumulative reward. The agent can learn this policy by interacting with the environment and observing rewards, which is called model-free RL. Some agents also learn a model of their environment and learn a policy based on simulated environment states. This is called model-based RL (Moerland et al., 2023).

However, a significant hurdle encountered by RL algorithms is the lack of sample efficiency (Micheli et al., 2023). The demand for extensive interactions with the environment to learn an effective policy can be prohibitive in many real-world applications (Yampolskiy, 2018). In response to this challenge, deep reinforcement learning (DRL) has emerged as a promising solution. DRL leverages neural networks to represent and approximate complex policies and value functions, allowing it to tackle a wide array of problems and environments effectively.

One such approach is the concept of "world models". World models aim to create a simulated environment within which the agent can generate an infinite amount of training data, thereby reducing the need for extensive interactions with the real environment (Ha & Schmidhuber, 2018). However, a key prerequisite for world models is the construction of precise models of the environment, a topic that has garnered substantial research attention (Hafner et al., 2022b; 2020; Kaiser et al., 2019). The idea of learning models of the agent's environment has been around for a long time (Nguyen & Widrow, 1990; Schmidhuber, 1990; Jordan & Rumelhart, 1992). After being popularized by Ha & Schmidhuber (2018), world models have since evolved and diversified to address the sample efficiency problem more effectively. Most prominently, the DreamerV1-3 models (Hafner et al., 2020; 2022c; 2023) have achieved state-of-the-art results in multiple benchmarks such as Atari100k (Bellemare et al., 2013) or Minecraft (Kanitscheider et al., 2021). These models use a recurrent state space

model (RSSM) (Hafner et al., 2022b) to learn a latent representation of the environment. The agent then uses this latent representation to train in imagination. Recently, Transformers have gained popularity as backbones for world models, due to their ability to capture complex dependencies in data. We will discuss some approaches using Transformer-based architectures in Appendix B. One of the major drawbacks of those architectures is the inherent lack of runtime efficiency of the attention mechanism used in Transformers. Recently proposed structured state space sequence (S4) models (Gu et al., 2022) show comparable or superior performance in a wide range of tasks while being more runtime efficient than Transformer-based models, which makes them a promising alternative (Deng et al., 2023; Lu et al., 2023).

Another avenue for improving RL sample efficiency is hierarchical RL (HRL) (Dayan & Hinton, 1992; Parr & Russell, 1997; Sutton et al., 1999). This approach operates at different time scales, allowing the agent to learn and make decisions across multiple levels of abstraction. The idea is that higher level policies divide the environment task into smaller subtasks or subgoals (also commonly called skills). The lower level policy is then rewarded for fulfilling these subgoals and is thus guided to fulfill the overall environment task. This approach has been shown to be effective in a variety of tasks (Hafner et al., 2022a; Nachum et al., 2018; Jiang et al., 2019; Nachum et al., 2019a). Hafner et al. (2022a) proposes an HRL approach that builds on DreamerV2 (Hafner et al., 2022c) and achieves superior results in the Atari100k benchmark.

Finding the right experience replay buffer sampling strategy is key for many RL algorithms, as it has a great influence on the final performance of the agent (Fedus et al., 2020; D'Oro et al., 2022). Li et al. (2023) introduce a time balanced replay dataset which empirically boosted the performance of their imagination based RL agent. However, this replay procedure relies on recomputing all probabilities in $O(n)$ at each iteration, which reduces its applicability for other approaches.

LeCun (2022) theorizes that an HRL agent that learns a hierarchy of world models and features intrinsic motivation to guide exploration could potentially achieve human-level performance in a wide range of tasks. Nachum et al. (2019c) and Aubret et al. (2023) provide further reasoning that combining HRL with other successful approaches such as world models could lead to a significant improvement in sample efficiency. Motivated by these findings, we propose HIEROS, a multilevel HRL agent that learns a hierarchy of world models, which use S5 layers to predict next world states. Specifically, our contributions are as follows:

- We propose HIEROS, an HRL agent that learns a hierarchy of world models, and builds upon the Director (Hafner et al., 2022a) and on the DreamerV3 (Hafner et al., 2023) architecture.
- Furthermore, we propose S5WM, a world model that uses S5 layers to predict next world states, which has several beneficial properties compared to the RSSM used in DreamerV3 (Hafner et al., 2023) and several proposed Transformer-based alternatives (Micheli et al., 2023; Robine et al., 2023; Chen et al., 2022) as well as a recently proposed S4WM (Deng et al., 2023).
- We derive efficient time-balanced sampling (ETBS) for experience dataset sampling from the time-balanced sampling method proposed by Robine et al. (2023) with a sampling time complexity of $O(1)$.
- We show that HIEROS achieves a new state-of-the-art mean and median normalized human score in the Atari100k benchmark (Bellemare et al., 2013).
- We conduct a thorough ablation study showing the influence of the different components of HIEROS (e.g., world model choice, hierarchy depth, sampling procedure).

We describe our method in detail in Section 2. In Section 3, we evaluate HIEROS on the Atari100k benchmark (Bellemare et al., 2013) and analyze our results. Concluding remarks and ideas for future work are given in the final Section 4. We provide a section explaining some background concepts of the topic in Appendix A, an additional comparison with related work in Appendix B and a range of ablation studies in Appendix G.

## 2 METHODOLOGY

In the context of RL, an agent interacts with an environment at discrete time steps, denoted as $t$. For the Atari 100K benchmark, for instance, where the environment represents a game like Pong,

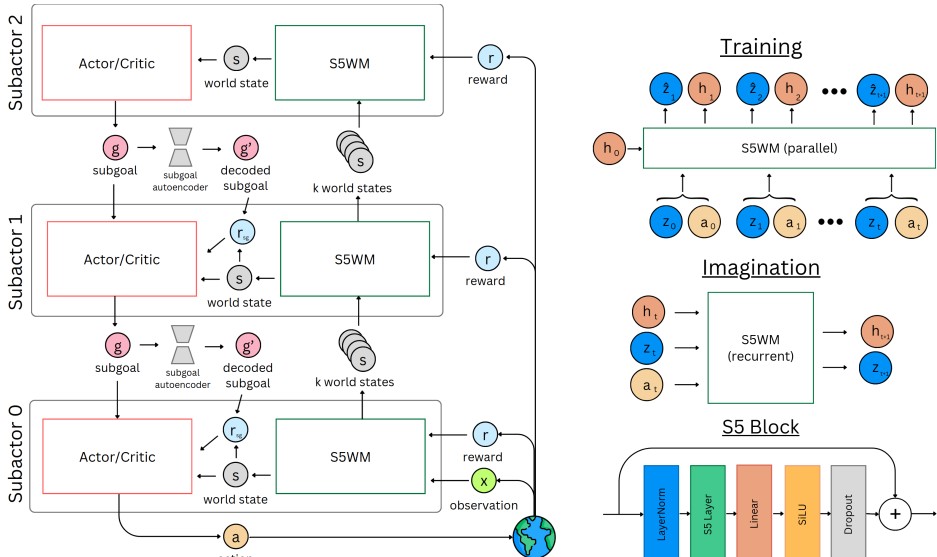

Figure 1: On the left: Hierarchical subactor structure of HIEROS. Each layer of the hierarchy learns its own latent state world model and interacts with the other layers via subgoal proposal. The action outputs of each actor/critic is the subgoal input of the next lower layer. The output of the lowest level actor/critic is the actual action in the real environment. On the right: Training and imagination procedure of the S5WM. HIEROS uses a stack of S5 blocks with their architecture shown above.

the agent's interaction involves selecting an action $a$ at time $t$ within the game, similar to making in-game moves or pressing buttons. Subsequently, the agent receives an observation $o$ and a reward $r$ from the environment. In the case of Pong, $o$ typically takes the form of a pixel image capturing the game's visual state, while $r$ represents the points earned as a result of the agent's actions. The agent's primary goal is to learn an optimal policy $\pi$, guiding its interactions with the environment, with the overarching objective of maximizing the expected cumulative reward, expressed as $\mathbb{E}\left[\sum_{t=0}^{\infty} \gamma^t r_t\right]$, where $\gamma < 1$ represents the discount factor and $r_t$ stands for the reward at time step $t$.

In this section we introduce HIERarchical imagination On Structured state space sequence world models (HIEROS), a hierarchical model-based RL agent, that learns on trajectories generated by a Simplified Structured State Space Sequence (S5) model (Smith et al., 2023). We mainly base our approach upon DreamerV3 (Hafner et al., 2023) and Director (Hafner et al., 2022a). In the following section, we propose two major changes to the DreamerV3 architecture. First, we describe a hierarchical policy, where each abstraction layer learns its own S5 world model and an actor-critic. Second, we replace the world model with a world model based on S5 layers. We also introduce efficient time-based sampling (ETBS) method for true uniform sampling over the experience dataset with $O(1)$ time complexity.

## 2.1 MULTILAYERED HIERARCHICAL IMAGINATION

HIEROS employs a goal conditioned hierarchical policy. Each abstraction layer learns its own S5 world model, an actor-critic and a subgoal autoencoder. We give some background on hierarchical reinforcement learning in Appendix A.2. The overall design of the subgoal proposal and the intrinsic reward computation is similar to the Director architecture (Hafner et al., 2022b), which successfully implements a hierarchical policy on the basis of DreamerV2. In contrast to Director and many other works, our architecture can easily be scaled to multiple hierarchy levels instead of being only two leveled (Hafner et al., 2022a; Nachum et al., 2018; Jiang et al., 2019; Nair & Finn, 2019).

We show the hierarchic structure and the interaction between the subactors in Figure 1. Each subactor $i$ consists of three modules: the world model ($w_\theta^i$), an actor-critic component ($\pi_\phi^i$), and a subgoal autoencoder ($g_\psi^i$). The world model is used to imagine trajectories for the actor-critic to train on. The subgoal autoencoder's task is to compress and subsequently decompress states within the world model. The decoder is used to decode the subgoals proposed by higher level layers and compute a

subgoal reward $r_g$. It is also used to compute novelty rewards $r_{nov}$. Only the lowest layer (subactor 0) interacts directly with the environment. The higher layers receive $k$ consecutive world model states from the lower level as observations, train their world model on these states and generate subgoals $g_i$. Director, in contrast, provides the higher level policy only with every $k$-th world state. We show the effect of this adjustment in Appendix G.5.

The subgoals represent world states that the lower layer is tasked to achieve and are kept constant for the next $k$ steps of the lower layer. So the higher levels can only update their proposed subgoal every $k$ steps of the next lower layer, so subactor 2 in Figure 1 can update its proposed subgoal every $k^2$ environment steps. The actor-critic component is trained on imagined rollouts of the word model. The reward $r$ is a mixture of extrinsic rewards $r_{extr}$ (i.e. observed rewards directly from the environment), subgoal rewards $r_g$, and novelty rewards $r_{nov}$:

$$r = r_{extr} + w_g \cdot r_g + w_{nov} \cdot r_{nov} \tag{1}$$

Here, $w_g$ and $w_{nov}$ are hyperparameters that control the influence of the subgoal and novelty rewards on the overall reward. The subgoal and novelty rewards $r_g$ and $r_{nov}$ are computed as follows:

$$r_{nov} = ||h_t - g_\psi^i(h_t)||_2 \qquad\qquad r_g = \frac{g_t^T \cdot h_t}{\max(||g_t||, ||h_t||)} \tag{2}$$

with $h_t$ being the deterministic world model state, $g_\psi^i$ the subgoal autoencoder of subactor $i$, $g_t$ the subgoal and $|| \cdot ||_2$ being the L2 norm. Since the subgoal autoencoder is trained to compress and decompress the world model state, it is able to model the distribution of the observed model states. This allows using its reconstruction error on the current world model state as a novelty reward $r_{nov}$. The subgoal reward is computed using the cosine max similarity method between the world model state and the decompressed subgoal, which was proposed by Hafner et al. (2022a).

The actor learns the policy:

$$a_t \sim \pi_\phi^i((h_t, z_t), g_t, r_t, c_t, H(p_\theta(z_t|m_t))) \tag{3}$$

with $h_t$ and $z_t$ representing the current model state, $g_t$ being the subgoal, $r_t$ being the reward, $c_t$ being the continue signal, and $H(p_\theta(z_t|m_t))$ being the entropy of the latent state distribution. The entropy term is used to encourage exploration and make the actor aware of states with high uncertainty. These additional inputs to the actor network are motivated by findings of Robine et al. (2023) which show, that providing the predicted reward as input improves the learned policy. The actor is trained using the REINFORCE algorithm (Williams, 1992). In the case of a higher level subactor, $a_t$ is the subgoal $g_t$ for the next lower layer. The actor-critic trains three separate value networks for each reward term to predict their future mean return. The subgoal autoencoder $g_\psi^i$ is composed of an encoder and a decoder:

$$g_\psi^i \text{ Encoder: } g_t \sim p_\psi(g_t|h_t) \qquad\qquad g_\psi^i \text{ Decoder: } h_t \approx f\psi(g_t) \tag{4}$$

Only the deterministic part of the model state $h_t$ is used as input for the subgoal autoencoder, as the stochastic part $z_t$ is less controllable by the lower level actor and thus would make the subgoal less achievable. Hafner et al. (2022a) found that the full latent state space of the lower level world model as action space would constitute a high dimensional continuous control problem for the higher level actor, which is hard for a policy to optimize on. Instead, the $g_\psi^i$ compresses the latent state into a discrete subgoal space of 8 categorical vectors of size 8. This compressed action space is much easier to navigate for the subgoal proposing policy than the continuous world state. Their worker policy takes the decompressed goal as input, while for HIEROS we found that the subactors achieved better subgoal completion and overall higher rewards being trained on the compressed subgoals. The subgoal autoencoder is trained using a variational loss:

$$\mathcal{L}_\psi = || f_\psi(z) - h_t ||_2 + \beta \cdot KL [ p_\psi(g_t | h_t) || q(g_t) ] \tag{5}$$

$z$ is sampled from the encoder distribution $z \sim p_\psi(g_t|h_t)$ and $q(g_t)$ is a uniform prior. $\beta$ is a hyperparameter that controls the influence of the KL divergence on the overall loss. All hyperparameters used in our experiments can be found in Appendix E.

The subgoals of all hierarchy layers can be decoded into the original image space of the game. This allows to visualize which subgoals the agent is trying to achieve, which makes the actions taken by HIEROS explainable. We show some examples in Appendix D.

## 2.2 S5-based World Model (S5WM)

The world model in HIEROS is responsible for imagining a trajectory of future world states. For this task, we use a similar architecture as DreamerV3 while swapping out the RNN-based sequence model for an S5-based sequence model. We give some background on world models and S4/S5 layers in Appendices A.1 and A.3. Our world model consists of the networks:

$$
\begin{aligned}
\text{Sequence model:} \quad & (m_t, h_t) = f_\theta(h_{t-1}, z_{t-1}, a_{t-1}) \\
\text{Encoder:} \quad & z_t \sim q_\theta(z_t \mid o_t) \\
\text{Dynamics predictor:} \quad & \hat{z}_t \sim p_\theta(\hat{z}_t \mid m_t) \\
\text{Reward predictor:} \quad & \hat{r}_t \sim p_\theta(\hat{r}_t \mid h_t, z_t) \\
\text{Continue predictor:} \quad & \hat{c}_t \sim p_\theta(\hat{c}_t \mid h_t, z_t) \\
\text{Decoder:} \quad & \hat{o}_t \sim p_\theta(\hat{o}_t \mid h_t, z_t)
\end{aligned}
$$

With $o_t$ being the observation at time step $t$, $m_t$ the output of the sequence model, and $h_t$ the internal state of the S5 layers in the sequence model, which we use as the deterministic part of the world state. $z_t$ is the stochastic part of the latent world state, $a_t$ the action taken, $\hat{z}_t$ the predicted latent state, $\hat{r}_t$ the predicted extrinsic reward, $\hat{c}_t$ the predicted continue signal and $\hat{o}_t$ the decoded observation. Figure 1 shows the overall structure of the world model.

During training, the S5WM takes in a sequence of observations $o_0 \cdots o_t$ and actions $a_0 \cdots a_{t-1}$. The encoder computes the posterior $z_t$ from the observations $o_t$ in parallel. The sequence model predicts the next output $m_t$ and deterministic state $h_t$. The dynamics predictor uses the sequence output $m_t$ to predict the stochastic prior $\hat{z}_t$. The posterior $z_t$ then constitutes the stochastic state with knowledge of the actual observation $o_t$ and the prior $\hat{z}_t$ constitutes the stochastic state solely predicted from the deterministic sequence model output. During imagination, the actor only has access to the prior $\hat{z}$.

The RSSM encoder in DreamerV3 computes the posterior from both $o_t$ and the deterministic model state $h_t$, which makes the parallel computation of $z_t$ impossible. However, Chen et al. (2022) show, that accurate representation can also be learned by making the posterior $z_t$ independent of $h_t$ and only predict the distribution $p(z_t|o_t)$ which is why we can safely use this for HIEROS as well.

As explained in Appendix A.3, S5 and S4 layers can be used both for sequence modelling in parallel and iterative autoregressive next state prediction. This makes them more efficient than RNNs for training the model and more efficient than Transformer-based models for imagination. S5 layers are advantageous over S4 layers as they can model longer term dependencies and are more efficient in terms of memory usage. Also, the latent state $x_t$ of S5 layers are directly accessible, which allows HIEROS using the recurrent model state as the deterministic part of the world state $h_t$. Other approaches use the sequence model output as deterministic world model states (Chen et al., 2022; Deng et al., 2023; Micheli et al., 2023), but this leads to worse results in our experiments (Appendix G.2).

We use a modified version of an S5 layer for training an RL agent, which was proposed by Lu et al. (2023). They introduce a reset mechanism, which allows setting the internal state $h_t$ to its initial value $h_0$ during the parallel sequence prediction by also passing the continue signal $c_0 \cdots c_n$ to the sequence model. This allows the actor to train on trajectories that span multiple episodes without leaking information from the end of one episode to the beginning of the next. We employ the same mechanism for our S5WM, as trajectories spanning episode borders are commonly encountered during training. It is unclear if the S4WM proposed in Deng et al. (2023) faced the same challenge and if and how they solved it.

The sequence model of our S5WM uses a stack of multiple S5 blocks, as depicted in Figure 1. The design of this block is inspired by the deep sequence model architecture proposed by Smith et al. (2023) but we selected a different norm layer, dropout and activation function. The number of blocks used is listed in Appendix E. All parts of S5WM are optimized jointly using a loss function that follows the loss function of DreamerV3:

$$
\mathcal{L}(\theta) = \mathcal{L}_{pred}(\theta) + \alpha_{dyn} \cdot \mathcal{L}_{dyn}(\theta) + \alpha_{rep} \cdot \mathcal{L}_{rep}(\theta) \tag{6}
$$

$$
\mathcal{L}_{pred}(\theta) = -\ln(p_\theta(r_t \mid h_t, z_t)) - \ln(p_\theta(c_t \mid h_t, z_t)) - \ln(p_\theta(o_t \mid h_t, z_t)) \tag{7}
$$

$$
\mathcal{L}_{dyn}(\theta) = \max(1, \text{KL}[\text{sg}(q_\theta(z_t \mid o_t)) \,\|\, p_\theta(z_t \mid m_t)]) \tag{8}
$$

$$
\mathcal{L}_{rep}(\theta) = \max(1, \text{KL}[q_\theta(z_t \mid o_t) \,\|\, \text{sg}(p_\theta(z_t \mid m_t))]) \tag{9}
$$

For $\mathcal{L}_{dyn}$ and $\mathcal{L}_{rep}$ we use the method of free bits introduced by Kingma et al. (2016) and used in DreamerV3 (Hafner et al., 2023) to prevent the dynamics and representations from collapsing into easily predictable distributions. $\alpha_{dyn}$ and $\alpha_{rep}$ are hyperparameters that control the influence of the dynamics and representation loss.

## 2.3 Efficient Time-balanced Sampling

When interacting with the environment, HIEROS collects observations, actions, and rewards in an experience dataset. After a fixed number of interactions, trajectories are sampled from the dataset in order to train the world model, actor, and subgoal autoencoder. DreamerV3 (Hafner et al., 2023) uses a uniform sampling. This, however, leads to an oversampling of the older entries of the dataset, as the iterative uniform sampling can select these entries more often than newer ones. As explained in Section 1 Robine et al. (2023) solve this using a time-balanced replay buffer with a $O(n)$ sampling runtime complexity with $n$ being the size of the replay buffer.

We propose an efficient time-balanced sampling method (ETBS), which produces similar results with $O(1)$ time complexity. When iteratively adding elements to the experience replay dataset and afterward sampling uniformly, the expected number of times $N_{x_i}$ an element $x_i$ has been drawn after $n$ iterations is

$$\mathbb{E}(N_{x_i}) = \frac{1}{i} + \frac{1}{i+1} + \cdots + \frac{1}{n} = H_n - H_i \tag{10}$$

with $H_i$ being the $i$-th harmonic number. The probability of sampling $x_i$, $i = 1, ..., n$, is

$$p_i = \frac{1}{n} \cdot \mathbb{E}(N_{x_i}) = \frac{H_n - H_i}{n} \approx \frac{\ln(n) - \ln(i)}{n} \tag{11}$$

We use the approximation $H_x \approx \ln(x)$ to remove the need to compute harmonic values. The idea is, to compute the CDF of this probability distribution between 0 and $n$ to transform samples from the imbalanced distribution into uniform samples via probability integral transformation David & Johnson (1948). How we derive the CDF of this distribution is shown in Appendix H. With the CDF we compute the ETBS probabilities as follows:

$$p_{etbs}(x) = CDF(\,p(x)\,) \cdot \tau + p(x) \cdot (1 - \tau) \tag{12}$$

with $p$ being the original sampling distribution, $CDF$ being the CDF of the original distribution and $\tau$ being a temperature hyperparameter that controls the influence of the original distribution on the overall distribution. We set $\tau$ to 0.3 in our experiments. Empirically, a slight oversampling of earlier experiences seems to have a positive influence on the actor performance. The time complexity of this sampling method is $O(1)$, as the CDF can be precomputed and the sampling is done in constant time. For further details, see Appendix H.

## 3 Experiments

We evaluate the performance of HIEROS on the Atari100k test suite (Bellemare et al., 2013), which contains a wide range of games with different dynamics and objectives. In each of these environments, the agent is only allowed 100k interaction with the environment, which amounts to roughly 2 hours of total gameplay. We evaluate HIEROS on a subset of 25 of those games. We compare HIEROS to the following baselines: DreamerV3 (Hafner et al., 2023), IRIS (Micheli et al., 2023), TWM (Robine et al., 2023), and SimPLe (Kaiser et al., 2019). SimPLe trains a policy on the direct pixel input of the environment using PPO (Schulman et al., 2017), while TWM, IRIS and DreamerV3 train a policy on imagined trajectories of a world model. TWM and IRIS use a Transformer-based world model. IRIS, however, trains the agent not in the latent space of the world model but in the decoded state space of the environment, which is one of the main differences to other approaches that leverage a Transformer-based world model (e.g. TWM). Ye et al. (2021) propose EfficientZero, which holds the current absolute state of the art in the Atari100k test suite. However, this method relies on look-ahead search during policy inference and is thus not comparable to the other methods.

Comparing our S5WM against the S4WM proposed by Deng et al. (2023) would be interesting, as both models are based on structured state spaces. However, Deng et al. (2023) only report results on their proposed set of memory testing environments and their code base is not public yet. So we

| Task | Random | Human | SimPLe | TWM | IRIS | DreamerV3 | Hieros (ours) |
|---|---|---|---|---|---|---|---|
| Mean | 0 | 100 | 34 | 96 | 105 | 112 | **120** |
| Median | 0 | 100 | 11 | 50 | 29 | 49 | **56** |
| IQM | 0 | 100 | 13 | 46 | 50 | N/A | **53** |
| Optimality Gap | 100 | 0 | 73 | 52 | 51 | N/A | **49** |

Table 1: Aggregate scores of HIEROS and baselines on the Atari100k test suite. Higher scores for Mean, Median and IQM are better. For Optimality Gap, lower scores are better. DreamerV3 does not report the scores for IQM or Optimality Gap. We show the best results for each row in **bold** font.

are not able to compare our results directly to theirs. We do however compare our S5WM to the RSSM used in DreamerV3 in Section 3.2. The used computation resources, implementation details and a link to the source code can be found in appendix F. We use the hyperparameters as specified in Appendix E for all experiments.

## 3.1 RESULTS FOR THE ATARI100K TEST SUITE

All scores are the average of three runs with different seeds. To aggregate the results, we compute the normalized human score (Bellemare et al., 2013), which is defined as $(score_{agent} - score_{random})/(score_{human} - score_{random})$. We show the achieved aggregated mean and median normalized human score in Table 1. The full table with scores for all games can be found in Appendix C.

Our model achieves a new state of the art in regard to the mean and median normalized human score. We also achieve a new state of the art in regard to the achieved reward on 9 of the 25 games. Adhering to Agarwal et al. (2021), we also report the optimality gap and the interquartile mean (IQM) of the human normalized scores and achieve state-of-the-art results in both of those metrics. HIEROS outperforms the other approaches while having significant advantages with regard to runtime efficiency during training and inference, as well as resource demand. For training on a single Atari game for 100k steps, TWM takes roughly 0.8 days on a A100 GPU, while DreamerV3 takes 0.5 days and IRIS takes 7 days. Hieros takes roughly 14 hours $\approx 0.6$ days and is thus significantly faster than IRIS while being on par with DreamerV3 and TWM.

The most significant improvements were achieved in Frostbite, JamesBond, and PrivateEye, which are games that feature multiple levels with changing dynamics and reward distributions. In order for the S5WM to learn to simulate these levels, the actor needs to employ a sufficient exploration strategy to discover these levels. This shift in the state distribution poses a challenge for many imagination-based approaches (Micheli et al., 2023). HIEROS is able to overcome this challenge by using the proposed subgoals on different time scales to guide the agent towards the next level. We show in Appendix D different proposed subgoals which guide the lower level actor to finding the way to the next level by building the igloo in the upper right part of the image.

HIEROS seems to perform significantly worse than other approaches in Breakout or Pong, which feature no significant shift in states or dynamics. It seems as if the hierarchical structure makes it harder to grasp the relatively simple dynamics of these games, as the dynamics remain the same across all time abstractions. This is backed by our findings in Appendix G.3 that HIEROS with only one subactor performs significantly better on Breakout. We also show empirically in Section 3.2 that using the S5WM also seems to deteriorate the performance of HIEROS in those games compared to the RSSM used for DreamerV3. Figure 5 shows some proposed subgoals for Breakout. The subgoals seem only to propose to increase the level score, which is does not provide the lower level agent any indication on how to do so. So the lower level actor is not able to benefit from the hierarchical structure in these games.

Figure 2 shows the partial rewards $r_{extr}, r_{nov}$, and $r_{sg}$ for the lowest level subactor for Breakout and Krull, another game featuring multiple levels similar to Frostbite. As can be seen, the extrinsic rewards for Breakout are very sparse and do not provide any indication on how to increase the level score. So the subactor learns to follow the subgoals from the higher levels more closely instead, which in the case of Breakout or Pong does not lead to a better performance. In Krull however, the extrinsic rewards are more frequent and provide a better indication on how to increase the level

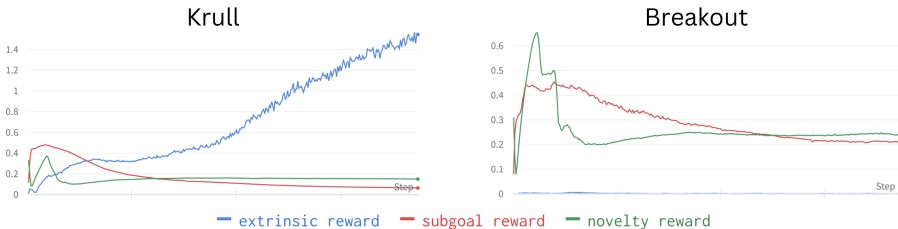

Figure 2: Extrinsic, subgoal, and novelty rewards for Krull (left window) and Breakout (right window) for the lowest level subactor.

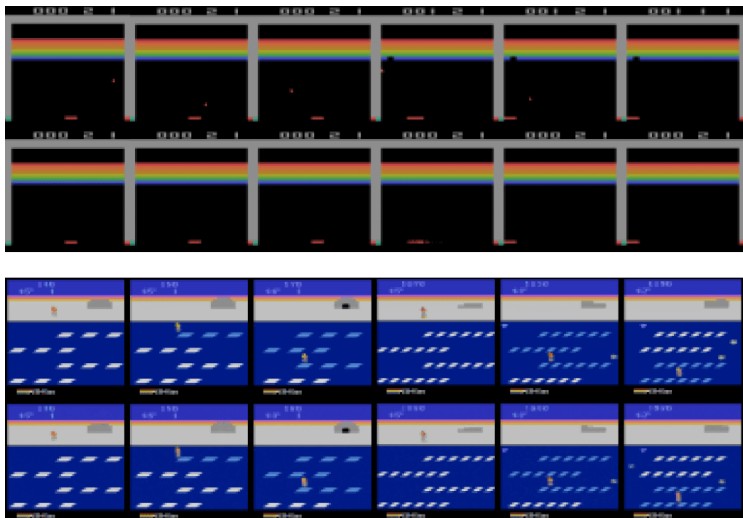

Figure 3: Trajectories for Breakout (top) and Frostbite (bottom). For each, the upper frame is the image observed in the environment and the lower frames are the imagined trajectories of the S5WM of the lowest level subactor.

score. So the subactor is able to learn to follow the subgoals from the higher levels more loosely, treating them more like a hint, and is able to achieve a better performance.

## 3.2 IMAGINED TRAJECTORIES

In Figure 3, we show an observed trajectory for the games Frostbite and Breakout alongside the imagined trajectories of the S5WM of the lowest level subactor. As can be seen, the world model is not able to predict the movement of the ball in breakout. This indicates that the model is not able to model the very small impact of the ball movement to the change in the image. We found that during random exploration at the beginning of the training the events where the ball bounces back from the moving plateau are very rare and most of the time the ball is lost before it can bounce back. This makes it difficult for the world model to learn the dynamics of the ball movement and their impact on the reward distribution. We make similar observations for the game Pong.

S4-based models are shown to perform worse than Transformer models on short term sequence modeling tasks (Zuo et al., 2022; Mehta et al., 2022; Dao et al., 2022) while excelling on long term modeling tasks. This might explain why our S5WM seems to perform worse on Breakout or Pong compared to Frostbite or Krull. Freeway poses a similar challenge, with sparse rewards that are only achieved after a complex series of environment interaction. However, unlike with Breakout and Pong, in Freeway HIEROS is able to profit from its hierarchical structure in order to guide exploration. An example of this can be seen in Figure 5, where the subgoals are able to guide the agent across the road. S5WM is able to accurately capture the multiple levels of Frostbite, despite only having access to train on these levels after discovering them, which usually happens after roughly 50k interactions. As the reaching of the next level is directly connected to a large increase

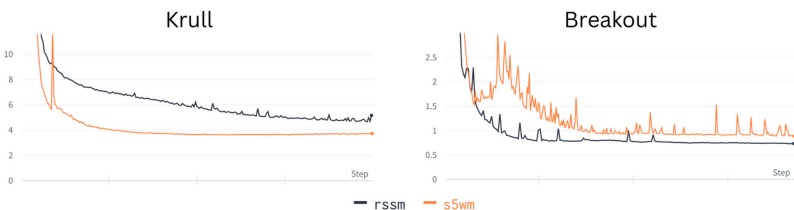

Figure 4: World model losses for the S5WM and RSSM for Krull and Breakout. The S5WM is able to achieve an overall lower world model loss compared to the RSSM for Krull, while those roles are reversed for Breakout.

in rewards, the S5WM is able to correctly predict the next level after only a few interactions. This is also reflected in the significantly higher reward achieved by HIEROS.

To directly compare the influence of our S5WM architecture to the RSSM used in DreamerV3, we replace the S5WM with an RSSM and train HIEROS on four different games. The results are shown in Appendix G.1. The RSSM underperforms compared to S5WM for Krull and Freeway, but can roughly match the performance of the S5WM for Breakout and Battle Zone. In Figure 4 we show the world model losses for S5WM and RSSM for Krull and Breakout. The S5WM is able to achieve an overall lower loss compared to the RSSM for Krull, while those roles are reversed for Breakout. Given that the absolute loss values for the more complex Krull are significantly higher than for Breakout, we conclude, that the more complex and also larger S5WM excels in environments where the dynamics are complex, and the input distribution experiences larger shifts, while the smaller RSSM is better suited for environments with simple dynamics and a stable input distribution.

As both models are trained with the same number of gradient updates, it makes sense that the larger model has difficulties matching the performance of the smaller model in non-shifting environments with simple dynamics. So a smaller S5WM might achieve better results for Breakout. Lu et al. (2023) and Deng et al. (2023) find that structured state space models generally surpass RNNs in terms of memory recollection and resilience to distribution shifts, which we can confirm with our results. Moreover, we perform further ablations in Appendix G.

## 4 CONCLUSION

In this paper, we introduce the HIEROS architecture, a multilayered goal conditioned hierarchical reinforcement learning (HRL) algorithm with hierarchical world models, an S5 layer-based world model (S5WM) and an efficient time-balanced sampling (ETBS) method which allows for a true uniform sampling over the experience dataset. We evaluate HIEROS on the Atari100k test suite (Bellemare et al., 2013) and achieve a new state of the art mean human normalized score for model-based RL agents without look-ahead search. The option to decode proposed subgoals gives some explainability to the actions taken by HIEROS. A deeper evaluation on which subgoals are proposed and how the lower level workers are able to achieve these subgoals is left for future research. The S5WM poses multiple improvements compared to RNN-based world models (Hafner et al., 2023) (i.e. efficiency during training, prediction accuracy during imagination) and Transformer-based world models (Chen et al., 2022; Robine et al., 2023; Micheli et al., 2023) (i.e. prediction accuracy and efficiency during imagination). A thorough comparison between our S5WM and the S4WM proposed by Deng et al. (2023) remains for future research. We delve deeper into further directions for future research in Appendix I.

We would also like to point out that the model-free SR-SPR recently proposed by D'Oro et al. (2022) achieves superior scores on the Atari100k benchmark by increasing the replay train ratio and regular resets of all model parameters. They tackle the often observed inability of RL agents to learn new behavior after having been already trained for some time in an environment. However, since this approach is not model based and does not train in imagination, we did not include it into our comparison. Nonetheless, scaling the train ratio and employing model resets for the actor/critic or the S5WM of HIEROS are promising directions for future research.

With our work, we hope to provide a new perspective on the field of HRL and to inspire future research in this field.

## REPRODUCIBILITY STATEMENT

We describe all important architectural and training details in Section 2 and provide the used hyperparameters in Appendix E. We provide the source code in the supplementary material and under the following link: `https://github.com/Snagnar/Hieros`. The material also gives an explanation on how to install and use the Atari100k benchmark for reproducing our results. The computational resources we used for our experiments are described in Appendix F.

## ETHICS STATEMENT

Autonomous agents pose many ethical concerns, as they are able to act in the real world and can cause harm to humans. In our work, we only use simulated environments and do not see any potential for misuse of our work. We propose a new world model architecture which could be used to train agents in imagination rather than in the real world. This could be used to train agents for real-world applications, such as autonomous driving, without the need to train them in the real world. This could reduce the risk of harm to humans and the environment.

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

## A BACKGROUND

### A.1 LEARNING A WORLD MODEL IN DREAMERV3

Our method is build on top of DreamerV3 (Hafner et al., 2023). DreamerV3 learns a world model in a latent space in order to increase efficiency:

$$(h_t, z_t) = \text{RSSM}(h_{t-1}, a_{t-1}, o_t) \tag{13}$$

with $h_t$ being the deterministic part of the latent state and $z_t$ being the stochastic part of the latent state. The world model is trained to predict the next latent state $(h_{t+1}, z_{t+1})$ given the world model state $(h_t, z_t)$ and the current action $a_t$. Specifically, the world model learns a distribution $p_\theta(z_t|h_t, o_t)$ predicting the stochastic state from the last deterministic state and an observation $o_t$ and another distribution $q_\theta(z_t|h_t)$ predicting the stochastic state purely from the last deterministic state. $q_\theta$ is used during imagination. It then predicts the reward $r$, the continue signal $c$ and the decoded observation $o$ given the current world state, which enables DreamerV3 to train an actor critic entirely on imagined trajectories in latent space. The world model is trained with a loss function that is a weighted sum of the loss of the dynamic, observation, continue, and reward prediction.

The RSSM dynamic prediction model uses a GRU (Cho et al., 2014) to predict the next deterministic state $h_{t+1}$ and a discrete categorical distribution to sample $z_{t+1}$. There are several works that replace the GRU with different, more complex models (Chen et al., 2022; Robine et al., 2023; Deng et al., 2023).

For imagining the trajectories, the world model starts with an initial observation $o_0$ and an initial state $h_0$. It then computes the first world state $(h_0, z_0)$. The agent interacts with this model, which simulates the original environment:

$$a_t = \pi(h_t, z_t) \tag{14}$$
$$(h_{t+1}, z_{t+1}) = \text{RSSM}(h_t, a_t, z_{t+1}) \tag{15}$$

### A.2 HIERARCHICAL REINFORCEMENT LEARNING

Hierarchical models have been shown to be a powerful tool in RL (Dayan & Hinton, 1992; Parr & Russell, 1997; Sutton et al., 1999). The idea is to break down a complex task into easily achievable subtasks. This subtask definition can be done manually (Tessler et al., 2017) or automatically (Li et al., 2022; Nair & Finn, 2019; Kujanpää et al., 2023). This typically involves learning a high level actor that works at larger timescale and a low level actor that executes proposed subgoals (Hafner et al., 2022a; Nachum et al., 2018; Jiang et al., 2019; Nachum et al., 2019a):

$$g_t = \pi_{high}(s_t) \tag{16}$$
$$a_t = \pi_{low}(s_t, g_t) \tag{17}$$

with $s_t$ being the current environment state, $g_t$ being the proposed subgoal and $a_t$ being the action taken by the low level actor. The low level actor is often encouraged to fulfill the subgoal by adding a subgoal reward $r_g$ to the extrinsic reward $r_{extr}$. This subgoal reward is often a function of the distance between the current state and the proposed subgoal (Hafner et al., 2022a; Nachum et al., 2019a).

### A.3 STRUCTURED STATE SPACE SEQUENCE MODELS

Structured state space sequence models (S4) were initially introduced by Gu et al. (2022) as a sequence modeling method that is able to achieve superior long-term memory tasks than Transformer-based models while having a lower runtime complexity ($O(n^2)$ for the attention mechanism of Transformers and $O(n \log n)$ for S4, $n$ being the sequence length). State Space models are composed of four matrices: $A$, $B$, $C$ and $D$. They take in a signal $u(t)$ and output a signal $y(t)$:

$$x(t+1) = Ax(t) + Bu(t) \tag{18}$$
$$y(t) = Cx(t) + Du(t) \tag{19}$$

with $x(t)$ being the state of the model at time $t$. The matrices $A$, $B$, $C$, and $D$ are learned during training. Gu et al. (2022) propose various techniques to increase stability, performance and training

speed of these models in order to model long sequences. They utilize special HIPPO initialization matrices for this. Another major advantage of S4 layers over Transformer models is, besides their better runtime efficiency, that they can be used both as a recurrent model, which allows for fast autoregressive single step prediction, and as a convolutional model, which allows for fast parallel sequence modelling. Smith et al. (2023) propose a simplified version of S4 layers (S5) that is able to achieve similar performance while being more stable and easier to train. Their version utilizes parallel scans and different matrix initialization in order to further boost the parallel sequence prediction and runtime performance. Lu et al. (2023) propose a resettable version of S5 layers, which allow resetting the internal state $x(t)$ during the parallel scans, in order to apply S5 layers in a RL setting where the state input sequence might span episode borders.

## B   FURTHER RELATED WORK

### B.1   HIERARCHICAL REINFORCEMENT LEARNING

Hierarchical reinforcement learning (HRL) is a field of RL that breaks down complex tasks in time and state abstracted subproblems on multiple time scales (Hutsebaut-Buysse et al., 2022; Sutton et al., 1999). This allows the agent to learn subtasks on different time scales and to reuse these subtasks in different contexts. This is especially useful in sparse reward environments, where the agent can learn subtasks that are easier to solve and then combine them to solve the overall task (Nair & Finn, 2019). Nachum et al. (2019c) show that one of the main benefits of HRL is improved exploration, more than inherent hierarchical structures of the problem task itself or larger model sizes. LeCun (2022) argue that HRL is a promising direction for future research in RL, as complex dependencies often are only discoverable by abstraction, as learned skills are often reusable in different contexts. Also, they show that HRL has deep rooting in human cognition.

Goal-conditioned HRL (Florensa et al., 2017; Nachum et al., 2018; 2019b;a) is a subfield of HRL that uses goal-conditioned policies to learn subtasks. The agent learns a policy that takes a goal as input and outputs actions that lead to the goal. The agent can then learn a policy that takes a goal as input and outputs a subgoal that leads to the goal. This allows the agent to learn subtasks on different time scales and to reuse these subtasks in different contexts. The higher order policy proposes subgoals in frequent intervals that the lower order policy has to fulfill, which is often incentivized by giving an intrinsic reward to the lower level policy (Hafner et al., 2022a; Rosete-Beas et al., 2023). Hafner et al. (2022a) combine a hierarchical policy with a world model, building on the DreamerV2 architecture (Hafner et al., 2022c). They show that the combination of a hierarchical policy and a world model outperforms the original DreamerV2 model on several tasks. The low level policy receives only subgoal rewards in this architecture, while the higher level policy receives the actual task reward. This is similar to our approach, but we use a different architecture for the world model, and we let the higher level policies learn a separate world model.

### B.2   WORLD MODELS

Environment interactions are typically expensive to train an RL agent. E.g., in robotic applications it is impossible to let the agent interact with the real environment, as this would be too expensive and potentially dangerous. Therefore, it is desirable to train the agent in a simulated environment (Ha & Schmidhuber, 2018; Poudel et al., 2022; Hafner et al., 2020; 2022c; 2023; 2022b). Ha & Schmidhuber (2018) introduced learning an RNN-based model of the environment and using this model to train the agent. This allows the agent to learn from simulated data, which is much cheaper than learning from real environment interactions and can in principle be generated in an infinite amount. Hafner et al. (2020) introduced Dreamer, a model-based RL agent that learns a world model based on PlaNet (Hafner et al., 2022b) and uses this world model to train the agent. PlaNet uses an RNN architecture which predicts the next world state from the last world state and the next action from the learned policy. It uses an RNN-based architecture. To increase computational efficiency, all learning is done in a compact latent state. They show that Dreamer outperforms state of the art model-free RL agents on several tasks.

Hafner et al. (2022c) introduced DreamerV2, an improved version of Dreamer featuring a discrete stochastic latent state. They show that DreamerV2 outperforms Dreamer on several tasks. With DreamerV3 (Hafner et al., 2023) they propose some additional improvements with which they were

able to solve the Minecraft Diamond challenge (Kanitscheider et al., 2021) without any pretraining. DreamerV3 still uses a PlaNet-based world model, but the model states are composites of a discrete valued stochastic and a continuous valued deterministic part.

Several authors propose improvements to this architecture, mostly by proposing improvements to the used world model. Chen et al. (2022) propose replacing the RNN with a Transformer model that takes in a context $(s_0, a_0), ..., (s_n, a_n)$ of state action pairs $(s_i, a_i)$ in order to compute the next world state. Robine et al. (2023) propose a similar architecture, but in contrast to the previous work, the Transformer model is not used during inference, which makes their model more computationally efficient. Deng et al. (2023) propose S4WM, utilizing S4 layers for the next state prediction. Since S4 layers can be used both for predicting sequences in parallel and predicting only the next value in an RNN like fashion, their model also proved to be more computationally efficient than the Transformer-based architectures and outperforms them in memorization capabilities.

## C  FULL ATARI100K BENCHMARK

| Task | Random | Human | SimPLe | TWM | IRIS | DreamerV3 | Hieros (ours) |
|---|---|---|---|---|---|---|---|
| Alien | 228 | 7 128 | 617 | 675 | 420 | **959** | 828 |
| Amidar | 6 | 1 720 | 74 | 123 | 143 | 139 | 127 |
| Assault | 222 | 742 | 527 | 683 | 1 524 | 706 | **1 764** |
| Asterix | 210 | 8 503 | **1 128** | 1 117 | 854 | 932 | 899 |
| BankHeist | 14 | 753 | 34 | 467 | 53 | **649** | 177 |
| Battle Zone | 2 360 | 37 188 | 4 031 | 5 068 | 13 074 | 12 250 | **15 140** |
| Boxing | 0 | 12 | 8 | 78 | 70 | **78** | 65 |
| Breakout | 2 | 30 | 16 | 20 | **84** | 31 | 10 |
| Chop.Command | 811 | 7 388 | 979 | **1 697** | 1 565 | 420 | 1 475 |
| CrazyClimber | 10 780 | 35 829 | 62 584 | 71 820 | 59 324 | **97 190** | 50 857 |
| DemonAttack | 152 | 1 971 | 208 | 350 | **2 034** | 303 | 1 480 |
| Freeway | 0 | 30 | 17 | 24 | **31** | 0 | **31** |
| Frostbite | 65 | 4 335 | 237 | 1 476 | 259 | 909 | **2 901** |
| Gopher | 258 | 2 412 | 597 | 1 675 | 2 236 | **3 730** | 1 473 |
| Hero | 1 027 | 30 826 | 2 657 | 7 254 | 7 037 | **11 161** | 7 890 |
| JamesBond | 29 | 303 | 100 | 362 | 463 | 445 | **939** |
| Kangaroo | 52 | 3 035 | 51 | 1 240 | 838 | 4 098 | **6 590** |
| Krull | 1 598 | 2 666 | 2 205 | 6 349 | 6 616 | 7 782 | **8 130** |
| KungFuMaster | 258 | 22 736 | 14 862 | **24 555** | 21 760 | 21 420 | 18 793 |
| Ms.Packman | 307 | 6 952 | 1 480 | 1 588 | 999 | 1 327 | **1 771** |
| Pong | -21 | 15 | 13 | **18** | 15 | **18** | 5 |
| PrivateEye | 25 | 69 571 | 35 | 86 | 100 | 882 | **1 507** |
| Qbert | 164 | 13 455 | 1 289 | 3 331 | 746 | **3 405** | 770 |
| RoadRunner | 12 | 7 845 | 5 641 | 9 109 | 9 615 | 15 565 | **16 950** |
| Seaquest | 68 | 42 055 | 683 | **774** | 661 | 618 | 560 |
| Mean | 0 | 100 | 34 | 96 | 105 | 112 | **120** |
| Median | 0 | 100 | 11 | 50 | 29 | 49 | **56** |
| IQM | 0 | 100 | 13 | 46 | 50 | N/A | **53** |
| Optimality Gap | 100 | 0 | 73 | 52 | 51 | N/A | **49** |

Table 2: Scores of HIEROS and baselines on the Atari100k test suite. Higher scores for Mean, Median and IQM are better. For Optimality Gap, lower scores are better. DreamerV3 does not report the scores for IQM or Optimality Gap. We show the best results for each row in **bold** font.

## D    VISUALIZATION OF PROPOSED SUBGOALS

Figure 5 shows the proposed subgoals for one observation in Frostbite and one observation in Breakout.

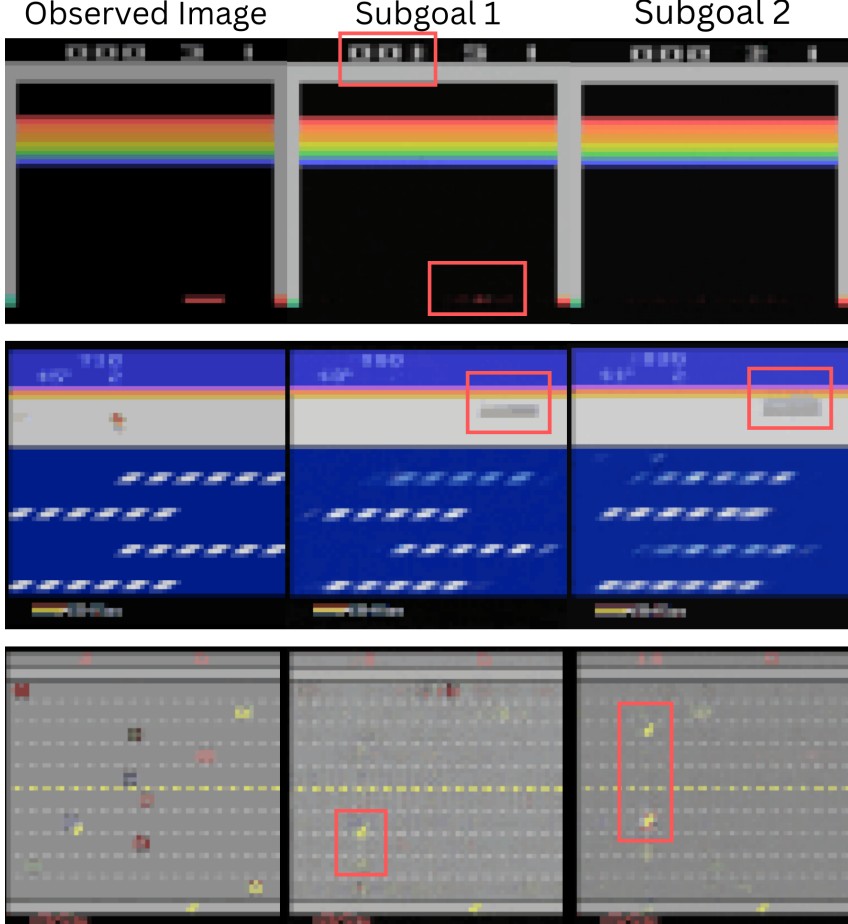

Figure 5: Proposed subgoals for Breakout (top row), Frostbite (middle row), and Freeway (bottom row). The left most frame is the original observation from the environment, and the following frames are the proposed subgoals from the higher level actor. For Breakout, the subgoals are only to increase the level score (marked with the red rectangles) and the ball is not simulated at all, while for Frostbite the subgoals guide the actor towards building up the igloo in the upper right part of the image in order to advance to the next level (red rectangles). For Freeway, which also features a single level and sparse rewards, the subgoals are much more meaningful than for Breakout and guide the actor to move across the road (red rectangles).

## E  HYPERPARAMETERS

| Training parameters: | |
| --- | --- |
| learning rate | $10^{-4}$ |
| weight decay | 0 |
| optimizer | AdamW |
| learning rate scheduler | None |
| warmup episodes | 1000 |
| ETBS temperature $\tau$ | 0.3 |
| batch size | 16 |
| trajectory length for training | 64 |
| imagination horizon | 16 |

| Hierarchy parameters: | |
| --- | --- |
| hierarchy layers | 3 |
| subgoal proposal intervals $k$ | 4 |
| extrinsic reward weight | 1 |
| subgoal reward weight | 0.3 |
| novelty reward weight | 0.1 |
| subgoal shape | 8x8 |

| World Model parameters: | |
| --- | --- |
| S5 model dimension | 256 |
| S5 state dimension | 128 |
| Number of HIPPO-N initialization blocks $J$ | 4 |
| Number of S5 blocks | 8 |
| dynamic loss weight $a_{dyn}$ | 0.5 |
| representation loss weight $a_{rep}$ | 0.1 |
| $h_t$ dimension | 256 |
| $z_t$ dimension | 32x32 |
| MLP units | 256 |
| $g_\psi$ KL loss weight $\beta$ | 0.5 |

| Total parameters | 37.1 M |
| --- | --- |

## F  COMPUTATIONAL RESOURCES AND IMPLEMENTATION DETAILS

In our experiments we use a machine with an NVIDIA A100 graphics card with 40 GB of VRAM, 8 CPU cores and 32 GB RAM. Training HIEROS on one Atari game for 100k steps took roughly 14 hours in our setup.

We base our implementation on the Pytorch implementation of DreamerV3 (NM512, 2023) and on the Pytorch implementation of the S5 layer (C2D, 2023). As this version does not implement the resettable version of S5 and Lu et al. (2023) do not provide an open source implementation of their method, we implemented the reset mechanism ourselves in the provided source code. The source code is publicly available under the following URL: `https://github.com/Snagnar/Hieros`

## G  ABLATIONS

In the following, we provide additional ablation studies exploring the effect of different components of HIEROS. We conduct all ablations on four games: (i) Krull, a game that features multiple levels, (ii) Breakout, a game with a single level and simple dynamics, (iii) Battle Zone, a game with a single level and complex dynamics and (iv) Freeway, a game with difficult exploration properties (Micheli et al., 2023). We use the same hyperparameters as described in Appendix E for all ablations. We show the collected reward of the lowest level subactor of HIEROS with S5WM and all hyperparameters as specified in Appendix E in orange and the collected reward of HIEROS with the ablation in

black. We use the same color scheme for all ablation studies, except those where the graphs contain more than two lines.

## G.1   S5WM VS. RSSM

In Section 3.2, we compare the model losses and partial rewards of HIEROS in the game of Krull and Breakout using either RSSMs or S5WMs as world models. In Figure 6, we compare the collected rewards of HIEROS using either RSSMs (black) or S5WMs (orange) for Krull, Breakout, Battle Zone, and Freeway against each other.

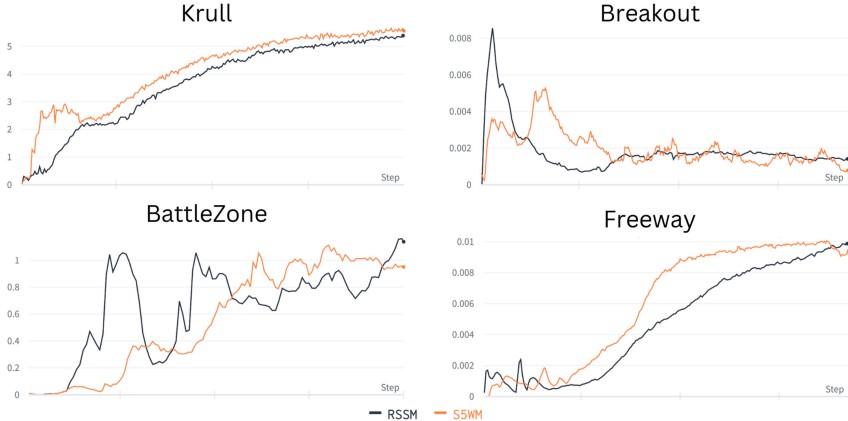

Figure 6: Collected rewards for Krull, Breakout, Battle Zone, and Freeway of the lowest level subactor for HIEROS with an RSSM and HIEROS with an S5WM. The RSSM is not able to achieve the same performance as the S5WM for Krull and Freeway, but can roughly match the performance of the S5WM for Breakout and Battle Zone.

## G.2   INTERNAL S5 LAYER STATE AS DETERMINISTIC WORLD STATE

In Section 2.2, we describe the S5WM, which uses the S5 layer to predict the next world state. In this section, we explore the effect of using the internal state $x_t$ of the stacked S5 layers of S5WM as the deterministic part of the latent state $h_t$ instead of the output of the S5 layers. Other comparable approaches that swap out the GRU in the RSSM with a sequence model usually use the output of the sequence model as $h_t$ (Chen et al., 2022; Micheli et al., 2023; Robine et al., 2023; Deng et al., 2023). So, testing this ablation provides valuable insight into how the learned world states can be enhanced in order to boost prediction performance. Figure 7 shows the influence of using the internal state as $h_t$ vs. using the output of the S5 layers as $h_t$ for HIEROS.

## G.3   HIERARCHY DEPTH

Here we show the effect of using different model hierarchy depths. We compare HIEROS with S5WM using a model hierarchy depth of 1, 2, 3, and 4. Figure 8 shows the collected reward of HIEROS with S5WM using a model hierarchy depth of 1 (light blue), 2 (black), 3 (orange), and 4 (green) for Krull, Breakout, Battle Zone, and Freeway. Most remarkably, we can see that HIEROS with just one layer achieves significantly better results than with two or more subactors. This indicates that a single layer algorithms, like DreamerV3, Iris, or TWM have a significant advantage over multi-layer algorithms, like HIEROS in games with no distribution shifts and easy to predict dynamics.

## G.4   UNIFORM VS. TIME-BALANCED REPLAY SAMPLING

In Section 2.3, we describe the efficient time-balanced sampling method. In this section, we compare the effect of using uniform sampling vs. our efficient time-balanced sampling for the experience dataset. Figure 9 shows the collected reward of HIEROS with S5WM using uniform sampling (orange) and time-balanced sampling (black) for Krull, Breakout, Battle Zone, and Freeway. As can

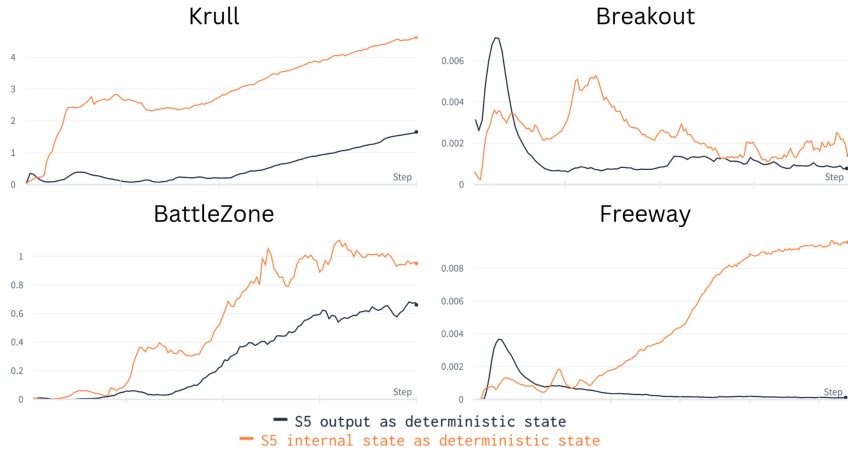

Figure 7: Comparison of the collected reward of HIEROS with S5WM using the internal state $x_t$ of the stacked S5 layers as the deterministic part of the latent state $h_t$ (orange) and using the output of the stacked S5 layers as $h_t$ (black) for Krull, Breakout, Battle Zone, and Freeway.

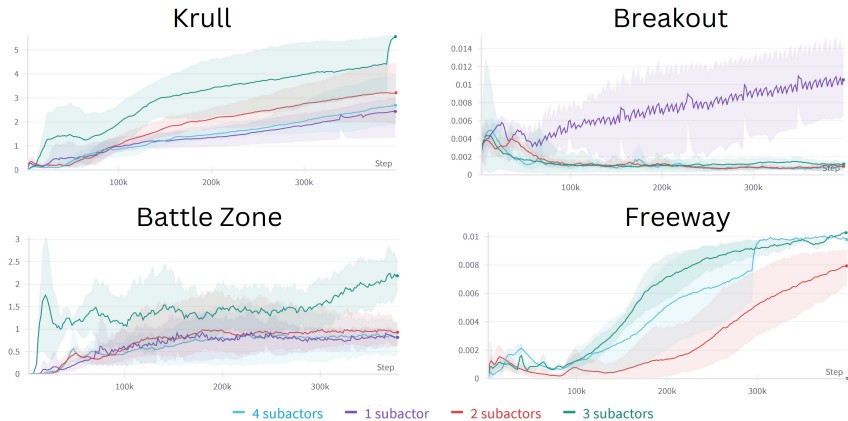

Figure 8: Comparison of the collected reward of HIEROS with S5WM using a model hierarchy depth of 1 (purple), 2 (red), 3 (green), and 4 (light blue) for Krull, Breakout, Battle Zone, and Freeway.

be seen, in the case of BattleZone and Breakout, the time balanced sampling did not provide a significant advantage, while it boosted the performance considerably for Krull and Freeway, the two games that display hierarchical challenges. This indicates, that in cases where the model hierarchy can contribute a lot to the overall performance, the actor is more sensitive to overfitting on older training data, containing subgoals produced by less trained higher level subactors.

### G.5 PROVIDING k WORLD STATES VS. ONLY THE k-TH WORLD STATE AS INPUT FOR THE HIGHER LEVEL WORLD MODEL

In Section 2.1, we describe how HIEROS provides $k$ consecutive world states of the lower level world model as input for the higher level subactor. However, many approaches such as Director (Hafner et al., 2022a) only provide the $k$-th world state as input for the higher level world model. In this section, we explore the effect of providing only the $k$-th world state as input for the higher level world model. Figure 10 shows the collected reward of HIEROS with S5WM using $k$ consecutive world states as input for the higher level world model (orange) and only the $k$-th world state as input for the higher level world model (black) for Krull, Breakout, Battle Zone, and Freeway.

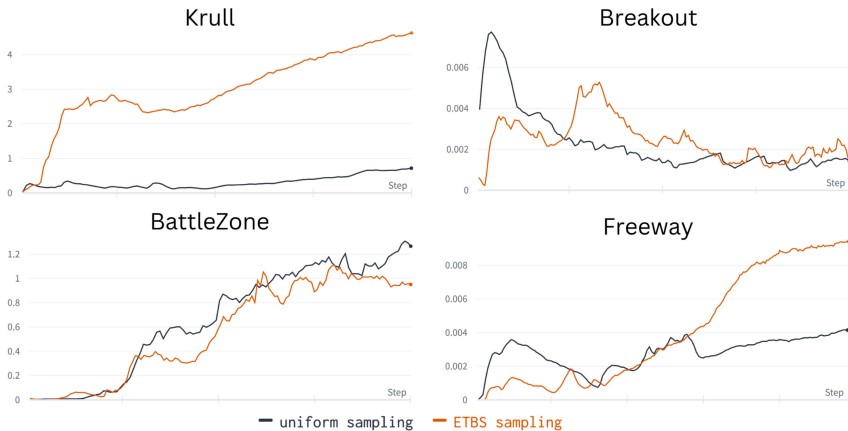

Figure 9: Comparison of the collected reward of HIEROS with S5WM using uniform sampling (orange) and time-balanced sampling (black) for Krull, Breakout, Battle Zone, and Freeway.

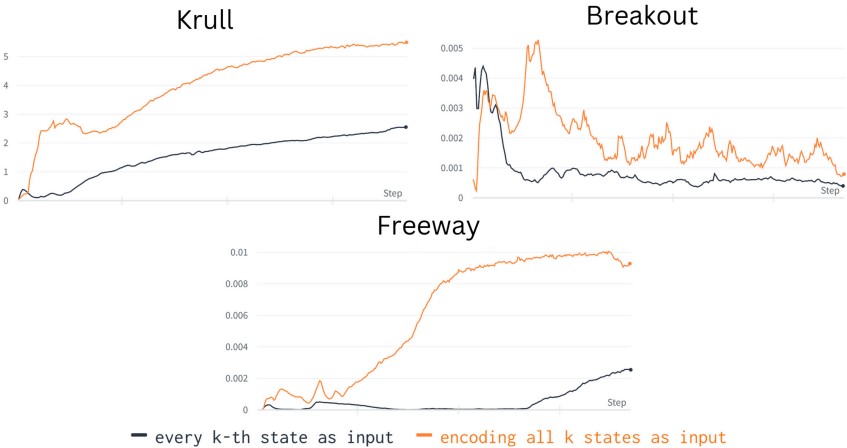

Figure 10: Comparison of the collected reward of HIEROS with S5WM using $k$ consecutive world states as input for the higher level world model (orange) and only the $k$-th world state as input for the higher level world model (black) for Krull, Breakout, Battle Zone, and Freeway.

### G.6 FURTHER ABLATIONS LEFT FOR FUTURE WORK

There are multiple other interesting directions for further ablation studies: One of the main novelties of HIEROS is its use of hierarchical world models. Architectures like Dreamer (Hafner et al., 2022a) however train both the higher and the lower level policy on the same world model, so exploring this might give valuable insights, how much the hierarchical world models contribute to the performance of HIEROS. Another interesting direction is to explore the effect of using differently sized subactors, with the lowest level subactor having the largest amount of trainable parameters and the higher levels having less and less trainable parameters. This could potentially lead to a more efficient use of the available parameters. The higher levels are trained fewer times and with fewer data than the lower levels, so smaller networks might speed up training in those cases.

For HIEROS we relied on the proven world model architecture used by the DreamerV3 model, which features a world state composed of a deterministic and a stochastic part. However, other approaches like IRIS (Micheli et al., 2023) do not rely on this stochastic world state and achieve comparable result. So exploring the effect of using only a deterministic world state might be interesting.

## H  CDF OF IMBALANCED REPLAY SAMPLING

In Section 2.3, we use the CDF of the skewed sampling distribution that arises when iteratively adding elements to a dataset and sampling uniformly from it. The distribution is defined as follows:

$$p(x) = \frac{H_n - H_x}{n} \approx \frac{\ln(n) - \ln(x)}{n} = \tilde{p}(x) \tag{20}$$

with $n$ being the size of the dataset. Since the inequality $\ln(x) < H_x < 1 + \ln(x)$ holds for all $x \geq 2$ we assume without loss of generality that $2 \leq x \leq n$. However, since $\tilde{p}(x)$ does not sum up to 1 over the interval $[2, n]$, we need to divide it by its integral:

$$p_s(x) = \frac{\tilde{p}(x)}{\int_2^n \tilde{p}(x)dx} = \frac{\frac{\ln(n)-\ln(x)}{n}}{\frac{-2\ln(n)+n+2\ln(2)-2}{n}} = \frac{\ln(n) - \ln(x)}{-2\ln(n) + n + 2\ln(2) - 2} \tag{21}$$

with $p_s$ being the approximate probability density function of the skewed sampling distribution. The CDF of this distribution is defined as follows:

$$CDF_s(x) = \int_2^x p_s(x)dx = P_s(x) - P_s(2) \tag{22}$$

with $P_s(x)$ being the antiderivative of $p_s(x)$:

$$P_s(x) = \int p_s(x)dx = \frac{x \cdot (\ln(x) - \ln(n) - 1)}{2\ln(n) - n - 2\ln(2) + 2} \tag{23}$$

With this, we can derive $CDF_s(x)$ in closed form:

$$CDF_s(x) = \frac{x \cdot (\ln(x) - \ln(n) - 1) + 2(\ln(n) - \ln(2) + 1)}{2\ln(n) - n - 2\ln(2) + 2} \tag{24}$$

This can be computed in $O(1)$ time and is therefore very efficient. With the $CDF_s(x)$ we can calculate the efficient time-balanced sampling distribution $p_{etbs}(x)$ as described in Section 2.3. It is also important to mention that we assume, that we only add one item to the dataset and then sample one time after each step. However, if we add $k$ items before sampling $s$ times, the expected number of draws for one element becomes $\mathbb{E}(N_{x_i}) = \frac{k \cdot (H_n - H_i)}{s \cdot n}$. This additional factor $\frac{k}{s}$ is canceled out when computing the probabilities for one element from the expected value, which is why we can ignore this in our computations. Figure 11 shows the sampling counts for different temperatures $\tau$ for ETBS.

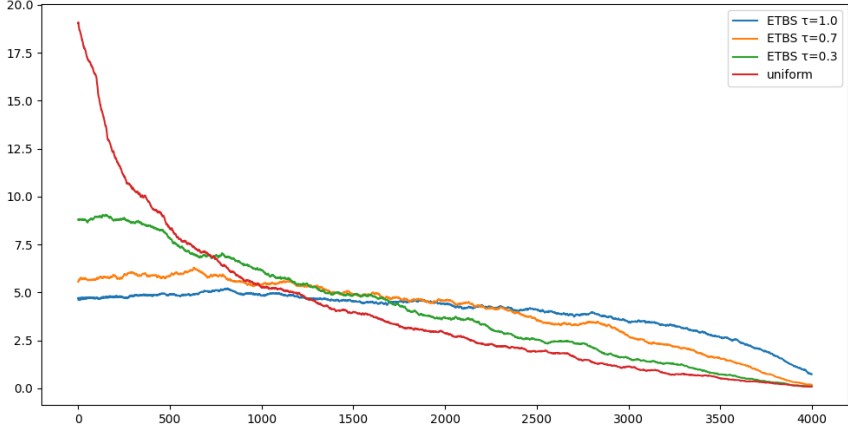

Figure 11: The number of times an entry $i$ is sampled in a dataset of final size 4000 with iteratively adding one element to the dataset and sampling over the dataset afterwards. Compared are the original uniform sampling method implemented in DreamerV3 (Hafner et al., 2023) and our proposed ETBS with different temperatures $\tau$. For our final experiments, we use a temperature of $\tau = 0.3$.

## I   FURTHER FUTURE WORK

Another possible future direction would be to use the S5-based actor network proposed by Lu et al. (2023) for HIEROS. Right now, the imagination procedure still relies on a single step prediction of the world model due to the single step architecture of the actor network. Using the S5-based actor network would allow for a multistep imagination procedure, which could further improve the performance of HIEROS. This would also open the door to more efficient look-ahead search methods.

Since S4/S5-based models and Transformer-based models show complementary strengths in regard to short and long-term memory recall, many hybrid models have been proposed (Dao et al., 2022; Zuo et al., 2022; Mehta et al., 2022; Gupta et al., 2022). Exploring these more universal models might also be a promising direction for future research.

LeCun (2022) describes a modular RL architecture, which combines HRL, world models, intrinsic motivation, and look-ahead planning in imagination as a potential candidate architecture for a true general intelligent agent. They propose learning a reconstruction free latent space to prevent a collapse of the learned representations, which is already explored in several works for RL (Okada & Taniguchi, 2021; Schwarzer et al., 2021). They also describe two modes of environment interaction: reactive (Mode 1) and using look-ahead search (Mode 2). HIEROS implements several parts of this architecture, namely the hierarchical structure, hierarchical world models and the intrinsic motivation. HIEROS, like most RL approaches, uses the reactive mode, while approaches like EfficientZero (Ye et al., 2021) could be interpreted as Mode 2 actors. Koul et al. (2020) implement a Monte Carlo Tree Search (MCTS) planning method in the imagination of a Dreamer world model. Implementing similar methods for the hierarchical structure of HIEROS, combined with the more efficient S5-based world model (potentially also an S5 based actor network) could yield a highly efficient planning agent capable of learning complex behavior in very dynamic and stochastic environments.

Since Figure 8 demonstrates that for some environments a deeper hierarchy can deteriorate performance, a possible future research could include an automatic scaling of the hierarchy depending on the current environment. E.g. if the accumulated reward stagnates and the agent cannot find a policy that further improves performance, the agent might automatically add another hierarchy layer, perform a model parameter reset and retrain on the collected experience dataset.

