# OpenReview forum: "Hieros: Hierarchical Imagination on Structured State Space Sequence World Models"
_ICLR.cc/2024/Conference — Submitted to ICLR 2024_

### Official Review · Reviewer_5ii1 · 2023-11-04

**Soundness:** 4 excellent
**Presentation:** 4 excellent
**Contribution:** 3 good
**Rating:** 8
**Confidence:** 4

**Summary:**

In this paper the authors introduce HIEROS, a hierarchical RL approach that learns a hierarchy of world models. These world models have a novel architecture based on S5 layers/blocks, referred to as S5WM. In addition, the authors employ a novel sampling strategy for training to ensure a true uniform strategy. The authors present each of these components and then include a number of experiments comparing HIEROS to relevant baselines on the Atari 100k benchmark. They find that their approach leads to a new SOTA on games with changing dynamics and reward distributions but that it struggled on simpler games.

**Strengths:**

This is a strong paper. The originality is perhaps the weakest part, largely made up of taking recent contributions (S5, sampling strategies, etc.) from several different fields. However, even with that there are significant developments made beyond these recent contributions. The quality is very high, with excellent descriptions of the approach, well-motivated experiments, and clear discussion of the results. The clarity could be improved, but arguments and discussions are still presented at a very high level. Finally, the significance is clear in HELIOS achieving SOTA in several games from the Atari 100k benchmark, though the performance on some simpler games is worrying.

**Weaknesses:**

As stated above, this is a strong paper. However, I identify two groups of weaknesses that could further improve the paper if addressed.

The first is in terms of the clarity. While overall the paper is very well-written, there are some issues. First, some core concepts are not clearly defined, it took several paragraphs before I clearly understood the nature of "subgoals", whereas they could have been defined when first introduced. Second, there are some minor language issues (e.g. "usurally"->"usually"). Third, there are some odd choices in terms of what should go into the appendix vs. the main body of the paper. I don't think Figures 2 or 4 add much to the authors' argument, and could be safely exchanged for say the experiment with reduced hierarchy depths. Fourth, there's some overstated claims. For example, I'm not sure that 0.6 days is on par with 0.5 hours.

The second are the results, specifically in terms of the worse performance on simpler games. This limits the potential significance of HELIOS, as choosing to deploy it requires a deep understanding of the dynamics of a potential environment. While the authors' discussion explaining this problem is excellent, the ablation of hierarchy depth gives an option for directly solving the problem. If HELIOS could automatically approximate an appropriate depth, it could in turn achieve better performance across more environments. This would also help improve the work's originality.

Again, as stated above, this is a strong paper with fairly minor groups of weaknesses.

**Questions:**

1. Why do the authors consider 0.6 days to be on par with 0.5 hours?
2. Did the authors explore automatically determining the hierarchy depth?

---

> ### Author Response · Authors · 2023-11-22
> **Response to Reviewer 5ii1**
>
> Thank you very much for your feedback. We have added a sentence in the caption of Fig. 1 that clarifies that subgoals are the action outputs of higher level actors and hope that this might make the concept of subgoals a bit clearer to understand. We also went through the paper again and fixed some typos that we apparently missed for our first submission. Thank you very much for pointing this out. We also fixed a mistake where we stated that DreamerV3 takes 0.5 hours for training on one Atari100k game, where it indeed takes 0.5 days which is why the runtimes of Hieros and DreamerV3 are indeed comparable.
>
> During our research we did not consider looking into ways to automatically scale the hierarchy depth depending on the game at hand. We did however experience that the overall performance decreases if layers are added later during training as opposed to starting with the full number of layers and training all in parallel. So if there is an approach to automatically grow the number of hierarchy layers, it would need to be applied after an initial data collection phase using e.g. a random policy. But the resuts might very well be better overall, as you pointed out, since the optimal hierarchy depth definitely is dependent on the game at hand. We added a paragraph discussing this in the future work section.
>
> Again, thank you very much for your review of our paper.

---

### Official Review · Reviewer_UXtg · 2023-11-05

**Soundness:** 2 fair
**Presentation:** 3 good
**Contribution:** 2 fair
**Rating:** 3
**Confidence:** 4

**Summary:**

This paper proposes a hierarchical model-based approach called HIEROS. It combines elements of several previous works, such as the hierarchical goal-conditioned policies of DreamerV3, the S5 layers of [Smith et al] and the prioritized sampling scheme of [Robine et al]. Putting this together, they get improved results over existing model-based approaches on the Atari100K benchmark for sample-efficient RL. Some ablations and analyses are performed, showing the model sets reasonable temporally-extended goals.

**Strengths:**

- The idea of hierarchical world models is fundamentally promising, and this paper appears to make some progress in that direction.
- The authors agree to release their code

**Weaknesses:**

- The algorithm for the most part combines existing algorithmic components (DreamerV3 hierarchy, S5 layers, time-balanced sampling). It does provide some improvement over existing model-based methods, but this doesn't feel like a fundamental advance.
- The paper does not provide very comprehensive ablation experiments. There are ablation experiments in Appendix F, but they are only on a handful of games (4), and it appears that only 1 seed is used. Therefore, it's hard to draw robust conclusions from these experiments given the high variance of RL experiments and the fact that there is also significant variance across different games. It would be helpful if these ablation experiments also reported mean/median/IQM using bootstrap-based confidence intervals, to determine to what extent the differences are significant or not.
- In the main results, although mean/median/IQM are reported using the rliable library (which is good), the confidence intervals are not reported - therefore, it is unclear if the differences of the proposed method compared to the others are significant or not. Please add these to the updated version.
- The paper does not include a comparison to [1], which showed the simple model-free methods are able to achieve very good performance on Atari100k provided the replay ratio is sufficiently increased and the policy parameters reset. In particular, that method seems to outperform the one proposed here (Mean: 1.27 vs. 1.20, Median: 0.68 vs. 0.56, IQM: 0.63 vs. 0.53), while being simpler. Therefore, the claims of achieving a new SOTA on Atari100k are not true.


[1] SAMPLE-EFFICIENT REINFORCEMENT LEARNINGBY BREAKING THE REPLAY RATIO BARRIER [D'Oro & Schwarzer et al, ICLR 2023]

**Questions:**

- the sentence "The agent can learn this policy by interacting with the environment and observing rewards, which is called model-free RL. The agent can also learn a model of the environment and use this model to plan actions. This is called model-based RL." isn't completely accurate. Model-based methods still can interact with the environment and learn by observing rewards, the difference with model-free methods is that they use a learned model as an intermediate step. This enables updating the policy (if there is one) with simulated rollouts through the model which does not cost any samples. Also, model-based algorithms do not necessarily plan in the sense of computing an action sequence each time they need to act. They can also learn a policy inside the model and use that to compute actions in a feedforward manner.
- The idea of learning world models with neural networks was _not_ initially proposed by Ha & Schmidhuber in 2018 as mentioned in the intro - this idea goes back at least to 1989 :) see [1, 2, 3] for some early works. There is also an extensive literature in model-based planning in the robotics literature.


[1] The Truck Backer-upper: An Example of Self-learning in Neural Networks [Nguyen and Widrow, 1989]

[2] An on-line algorithm for dynamic reinforcement learning and planning in reactive environments [Schmidhuber, 1990]

[3] Forward models: Supervised learning with a distal teacher [Jordan and Rumelhart, 1992]

---

> ### Author Response · Authors · 2023-11-22
> **Response to Reviewer UXtg**
>
> Thank you very dearly for your review of our work. First of all, we would like to point out that the main novelty of our paper is the implementation of a multileveled imagination based RL agent, which to our knowledge has not been achieved before us (especially the multileveled imagination, which is a significant advancement in comparison to Director, the work closest to ours). The other enhancements, S5WM and ETBS, are more of an incremental nature.
>
> Regarding our ablation studies, we are currently conducting more experiments with more seeds and will update the paper as soon as those are finished. Due to limited time/resources, we were not able to provide these during the review reply duration. We apologize for this. We are also rerunning parts of the main experiment, as we identified some errors in our implementation there. We will provide the updated metrics (including the requested confidence intervals) as soon as these experiments are finished. Since the ablation for the number of subactors is already finished, we included an updated version of that figure in the new version of the paper.
>
> We added a short discussion of the paper you mentioned in our conclusion. However, since this approach does not directly include imagination based RL agents, the grounds for comparison are rather limited. However, since the changes to the replay ratio and regular resets of the models seem to be universally applicable, we will evaluate if and how this can be included in Hieros. Thank you very much for pointing us to this paper.
>
> Regarding your questions:
> 1. The mentioned sentence indeed is a bit misleading. We corrected that part of the introduction to be more precise. Thanks for bringing this to our attention.
> 2. We changed the part of the introduction that imprecisely names Ha & Schmidthuber as the sole inventors of the concept of world models and cited the mentioned sources in order to give a more precise overview over the field.
>
> We hope we have addressed most of your points, and that you might consider changing your review.

---

### Official Review · Reviewer_EjYN · 2023-11-06

**Soundness:** 2 fair
**Presentation:** 1 poor
**Contribution:** 3 good
**Rating:** 3
**Confidence:** 3

**Summary:**

The authors propose Hieros, a hierarchical policy learning approach for solving reinforcement learning problems with world models. The idea of their work is that stacks of policies that generate subgoals can help improve the sample efficiency when solving a reinforcement learning problem. The author's contribution includes replacing the typical recurrent state space model in Dreamer with an S5 sequence model, which is helpful in other problem domains like sequence modeling. The authors also point out that sampling data to train world models is not uniform. They propose a novel sampling approach that is more computationally efficient compared to previous solutions to adjust sampling probabilities to correctly sample uniformly from collected demonstration data. The authors justify their framework on the Atari 100k benchmark. Their results suggest Herios is superior on average across the games, particularly in those with less stationary dynamics.

**Strengths:**

Based on the provided previous works, the author's research is the next natural step in improving RL algorithms that learn from world models. Verifying and testing novel architectures in a new domain – in this case, the S5 layer in the context of model-based reinforcement learning – is an essential step in verifying the generality of the benefits of such models. Considering hierarchical models for world models is also helpful, as this will likely help adapt policies to related problems to those seen during training.
The author's approach to address sampling to train world models is a valuable contribution. It can be easy to overlook the consequences of the computations in an algorithm, so finding a faster means to address sampling issues is a beneficial tool to introduce to the community. Conceptually, the paper has a promising motivation, and the authors have a good base for their work. We expand on this in the weaknesses section.

**Weaknesses:**

Although the motivations of the paper are sound, the paper needs more work before it is ready for acceptance. The overall structure of the article needs work, and the experiments should be more conclusive because of the empirical nature of the work. We expand on each point with what follows.
In terms of writing, the paper would benefit from including additional sections that focus on other details besides the author's work: a background section and a potentially related work section. The latter could be absorbed into the introduction, though. At the moment, many paragraphs include a discussion better suited as related work discussion: "Author A proposed this, Author B proposed this, Author C is similar to us but differs because we use X while they study Y." The authors currently abuse the appendix for crucial information to understand their work.
A background section is essential in the main paper as it is difficult to discern the author's novel contributions and previous work. While reading the methods section, we found it frustrating reading the discussion about what previous work A or previous work B had proposed as opposed to what the authors did differently. The methods section should only discuss what the authors did and only briefly mention related research if, for example, they apply a technique directly from a previous work (like using the S5 layer). For instance, it took us three read-throughs to realize the action outputs of the actor/critic models in higher layers were generating them. The current article technically discusses this point, but it is easy to miss. Removing all related work and background information discussion would make it more apparent in the methods sections. The authors should also consider annotating these contributions, more evident in Figure 1 so that one can spot the proposal at a glance.
The other area for improvement of the writing is treating the proposed sampling approach as a footnote. While reading the introduction, we were surprised to see a new sampling approach listed as a contribution. The intro never mentions sampling from a data buffer as a problem. We encourage the authors to integrate and discuss this sampling approach in the next version of their introduction section, which otherwise feels disjoint from the rest of the research.
As for the experiments, the more valuable contributions of the author's work are the ablation experiments in Appendix F because of the variety of novel components the authors consider. Knowing the benefit of each aspect of the author's contributions is more inciteful than just a table of numbers to get a sense of the sensitivity of these results. We suggest re-evaluating the priorities of which results to focus on in the main paper.
However, it is difficult to discern whether the author's components are necessary from these additional results, as several environments show no benefit to improving performance. The authors discuss why this is the case, but some of this content reads like testable hypotheses that they choose not to investigate. The most notable example was not including experiments comparing a smaller S5WM for the experiments discussed in Appendix F.1. Results we point out the authors discuss in the main paper.
One solution to add ablation results to the main paper is to reconsider the value of including the full Table 1 results. We suggest moving these to the appendix and only having the highlights from it in the main paper. E.g., results from the environments Heiros did poorly on, the ones it did best on, and the aggregate metrics. Modifying Table 1 could create additional space to have other experiment results or space to include crucial details in the main paper.
 Figure 3 is more appropriate for the appendix section because it justifies the poor performance of Pong. The figure could be a single sentence and elaborated on in the appendix. For example, "Heiros did poorly in pong, and in the appendix, we include results which show that this could be associated with the world models not reconstructing the ball."
The most concerning limitation of the author's results is excluding the S4WM as a baseline. We require further justification for why the authors did not compare against this model because the current rationale is insufficient. From the author's discussion, S4WM seems the most natural model to compare against. If the repository was going to eventually be made publicly available, in the reviewer's opinion, they had several alternatives:
1) Wait to submit their paper until after the code is released.
2) Contact the previous work's authors to request access to compare against their models. If they received no response, then this would be more understandable.
3) Adjust their experiments to enable them to compare to the previous work directly.
Alternative 1, we point out, would allow the authors to run more trials for each experimental configuration—another weakness in trusting the current experimental results.
The authors acknowledge their ablations are non-exhaustive, and for this reason, it might be better for them to simplify their model by removing any changes that need to be better justified from previous world model research. For example, In equations 8 and 9, the authors mention using Kingma et al. (2016) free-bit ideas, but this comes off as an arbitrary decision with no experimental justification provided. Proposing changes without strong reasons creates doubt about the crucial components of the author's work that lead to the performance improvements observed.
Overall, this reviewer feels the authors have all the pieces to verify and conduct the necessary experiments to justify the Heiros framework. Our opinion, though, is that additional results are needed to more thoroughly validate the system, which is a matter of time as opposed to adding further novel contributions.

**Questions:**

1.Are world models not a just form of model-based RL?
2.Is the use of ETBS not a 3rd component in your framework? You say you make two changes, but that one seems like a 3rd change.
   - Hierarchical policy
   - Using S5 layer
   - ETBS sampling
3.Are Director and DreamerV3 the same thing or not? In the introduction, the authors say they build off of the Director, and then in the methods section, they build off Dreamer-v3.
4. Is the S5 block shown in Figure 1 from previous work? So layer norm, S5, linear, SiLU dropout is just prior work & included for completeness as opposed to contribution the authors propose?
5. Equation 2: what's the point of normalizing by the magnitude of the larger vector in reward signal r_g?
6. How does the sub-goal predicting components distinguish learning differing sub-goals? If this is the case, it might be helpful to clarify how more variations of the same goal would be beneficial at deployment.
7. Did the authors try to contact the authors of Deng et al. 2023 to get access to their code base to compare to their results? If we have the paper wrong, then we mean the work that proposed S4WM world models instead, which was listed as important to compare against but not compared against.
8. Have the authors seen any previous works that discuss the issue pointed out in the first paragraph of section 3.2? We've heard this is a known issue when doing direct image reconstruction in pixel space for images. The usual problem is that MLE losses are not heavily concerned with minor details (e.g., the Ball in Pong) as opposed to the more significant information (background, the blocks, etc.). We apologize for not providing any citations.
9. How many trials did you conduct for the ablation experiments in the appendix? Why are there no error bars? Also, how many epochs or steps did these experiments run for? We could not see a number on the x-axis.
10.How did you choose Krull, breakout, Freeway, and Battle Zone as the ablation environments?
11. What was the author's conclusion from Figure 9 in the appendix? ETBS sampling is one of the main contributions of the paper, but it doesn't seem easy to discern if it's helpful from the provided plot in the appendix.

---

> ### Author Response · Authors · 2023-11-22
> **Response to Reviewer EjYN**
>
> Thank you very dearly for your constructive feedback on our paper. We have changed most of the points you mentioned, however, some need more time to be resolved.
> - We decided to keep the related work and background section in the appendix for now, as they are not part of our work and the space in the main paper is very limited due to ICLR restrictions. However, we expanded some paragraphs in the introduction to give some more insights on the related papers that we reference most in our paper.
> - We reduced the amount of discussion of other papers in the methodology section. However, we keep the parts that give at least some background on the methodology so that readers do not have to jump to the Appendix first.
> - We added a sentence in the caption of Fig 1 clarifying that the action outputs of higher level actor/critics are the subgoals for lower level actor critics.
> - We added a description of the problem we are trying to solve with ETBS in the Introduction.
> - We moved the full result table to the appendix, leaving only the aggregated results in the main paper.
> - Regarding Fig. 3, we are leaving this figure in the main paper for now, as it not only shows why Hieros performed worse on Pong but also the general ability of S5WM to (mostly) correctly simulate game states multiple steps into the future.
> - We contacted the authors of the S4WM paper requesting their code to run our own experiments but did not receive an answer. The code itself is also still not publicly available.
>
> Unfortunately, due to time and resource constraints, we couldn't complete all validation runs for our initial findings before the review deadline. We apologize for this and plan to update the paper with the missing results within approximately a week. However, we've already included the updated figure for the ablation of the number of subactors in the new version of our paper.
>
> Regarding your questions:
> 1. Yes, world models pose a form of model based RL, however, world models can be used to explicitly model action dependent world states multiple steps into the future, while the term model based RL refers more broadly to any approach that uses a model to predict future rewards and state transitions.
> 2. Yes, we corrected the statements in our paper that made it seem like ETBS is not part of our contribution.
> 3. Hieros is mainly built upon the DreamerV3 architecture, but borrows ideas from Director in order to build a hierarchy of models. We added clarification in the paper.
> 4. The S5 block is inspired by the sequence model design in [1] but we changed some of the layers as they empirically improved performance. We added clarification to the paper.
> 5. The max cosine distance used for s_g improves on the cosine distance by encouraging that the magnitude of both state vector and the goal vector should also be equal. It was introduced in the Director paper, where they add extensive ablation studies showing the superiority of max-cosine over other distance metrics for state and goals.
> 6. The action spaces of higher level subactors (i.e. the subgoal space) are very restricted. The subgoal consists of 8 discrete one-hot vectors of size 8, so the actor can choose 8 discrete actions per subgoal. These restrictions force the subgoal autoencoder to find robust representations. Adding more variations of the same goal highly increases the robustness of the subgoal space and should improve overall performance.
> 7. Yes, we did and received no answer yet.
> 8. We found some papers that use L1 loss instead of MLE (e.g. Micheli et al. [2]), however, we did not find a conclusive comparative analysis of which loss works better for learning environment representations. Using MLE for the decoder loss produced empirically better results for Hieros.
> 9. We conducted three runs for each ablation experiment. @e are running more experiments and also test a few other ablations (e.g. using a smaller S5WM as per your suggestion) but we cannot provide the results during the short review reply duration and will update the paper as soon as they are available.
> 10. Our conclusion is that using ETBS positively influences games with hierarchical structures (e.g., Krull), matching the performance of DreamerV3's time-imbalanced sampling in simpler games (e.g., Breakout). In Krull, early replay records with less trained subactors impact performance, mitigated by additional training on new data. In Breakout, the multilevel hierarchy has minimal impact, allowing training on less accurate subgoals without significant negative effects. A concise conclusion has been added to the ablation section.
>
> Again, thank you a lot for your constructive feedback. We hope we have addressed most of your points and would be very glad if you decided to adapt your initial review.
>
> [1] Smith, Jimmy TH, Andrew Warrington, and Scott W. Linderman. "Simplified state space layers for sequence modeling."
> [2] Micheli, Vincent, Eloi Alonso, and François Fleuret. "Transformers are sample efficient world models."

---

### Meta-Review · Area_Chair_VtaY · 2023-12-14

**Metareview:**

This paper presents a novel hierarchical RL algorithm that make use of world models and planning at multiple levels of abstraction.

The topic of the paper is timely and important.

The algorithm proposed combines several existing methods in the literature in a novel way, and the resulting algorithm is reasonable and interesting.

However, the reviewers highlighted two main points of concerns, which after carefully reading the paper, I share:
1) Experimental results are not fully convincing. More experiments should be conducted, and the results in the appendix should be moved to the main text as they are central to some of the claims.
2) The manuscript does a very poor job at contextualizing its contribution in the existing literature. Reading this paper, it would seem that model-based reinforcement learning was invented in 2018, thus erasing decades of work. In addition, presenting a related work section at the beginning of the manuscript is important to contextualize contributions. Instead, here the related work section is in the appendix.

**Justification For Why Not Higher Score:**

While the paper has potential, accepting the manuscript in its current state would lower the standard of ICLR and perpetuate disputable scientific practices (including not presenting fairly past literature, and not evaluating algorithms thoroughly)

**Justification For Why Not Lower Score:**

N/A

---

### Decision · Program_Chairs · 2024-01-16

Reject